# Bayesian regression explains how human participants handle parameter uncertainty

**Jannes Jegminat** [1,2]*, **Maya A. Jastrzębowska** [3,4], **Matthew V. Pachai** [3,5], **Michael H. Herzog**[3], **Jean-Pascal Pfister** [1,2]

**1** Department of Physiology, University of Bern, Bern, Switzerland, **2** Institute of Neuroinformatics and Neuroscience Center Zurich, ETH and the University of Zurich, Zurich, Switzerland, **3** Laboratory of Psychophysics (LPSY), Brain Mind Institute, School of Life Sciences, École Polytechnique Fédérale de Lausanne (EPFL), Lausanne, Switzerland, **4** Laboratory for Research in Neuroimaging (LREN), Department of Clinical Neuroscience, Lausanne University Hospital (CHUV) and University of Lausanne, Lausanne, Switzerland, **5** Department of Psychology, York University, North York, Canada

* jannes@ini.uzh.ch

**Data Availability Statement:** All relevant data are within the manuscript and its Supporting Information files.

## Abstract

Accumulating evidence indicates that the human brain copes with sensory uncertainty in accordance with Bayes' rule. However, it is unknown how humans make predictions when the generative model of the task at hand is described by uncertain parameters. Here, we tested whether and how humans take parameter uncertainty into account in a regression task. Participants extrapolated a parabola from a limited number of noisy points, shown on a computer screen. The quadratic parameter was drawn from a bimodal prior distribution. We tested whether human observers take full advantage of the given information, including the likelihood of the quadratic parameter value given the observed points and the quadratic parameter's prior distribution. We compared human performance with Bayesian regression, which is the (Bayes) optimal solution to this problem, and three sub-optimal models, which are simpler to compute. Our results show that, under our specific experimental conditions, humans behave in a way that is consistent with Bayesian regression. Moreover, our results support the hypothesis that humans generate responses in a manner consistent with probability matching rather than Bayesian decision theory.

## Author summary

How do humans make prediction when the critical factor that influences the quality of the prediction is hidden? Here, we address this question by conducting a simple psychophysical experiment in which participants had to extrapolate a parabola with an unknown quadratic parameter. We show that in this task, humans perform in a manner consistent with the mathematically optimal model, i.e., Bayesian regression.

## Introduction

The brain evolved in an environment that requires fast decisions to be made based on noisy, ambiguous and sparse sensory information, using noisy information processing and noisy effectors. Hence, decisions are typically made under substantial uncertainty. The main idea

**Funding:** This study has been supported by the Swiss National Science Foundation (SNF). In particular, J.J. and J.-P.P. have been supported by the SNF grants "Inference and Learning with Spiking Neurons"(PP00P3_150637, http://p3.snf.ch/Project-150637) and "Bayesian synapses" (31003A_175644, http://p3.snf.ch/Project-175644). M.A.J., M.P. and M.H.H. have been supported by the SNF grant 'Basics of visual processing: from elements to figures' (176153, http://p3.snf.ch/Project-176153). The funders had no role in study design, data collection and analysis, decision to publish, or preparation of the manuscript.

**Competing interests:** The authors have declared that no competing interests exist.

behind Bayesian brain hypothesis is that the brain uses the framework of Bayesian probabilistic computation to make optimal decisions in the presence of uncertainty [1–3]. Despite various counterexamples, e.g., [4], a large body of research has established that many aspects of cognition are indeed well described by Bayesian statistics. These include magnitude estimation [5], color discrimination [6], cue combination [7], cross-modal integration [8, 9], integration of prior knowledge [10, 11] and motor control [12–14].

Some experimental studies have considered more complex tasks, including visual search [15, 16], same-different discrimination [17] and change detection [18], but most can be cast into the problem of estimating a hidden quantity from sensory input. Much fewer experimental studies have been performed on regression tasks (but see [19] for an overview, and, e.g., [20–22]). In a regression task, the aim is to learn the mapping from a stimulus $x$ to an output $y$ after having been exposed to a training dataset $D = \{(x_i, y_i)\}_{i=1}^N$ of $N$ associations between stimulus $x_i$ and its corresponding $y_i$. Since the mapping from $x$ to $y$ can be probabilistic, the aim of regression is to find an expression for $p(y|x, D)$. Classification tasks, such as object recognition, or self-supervised tasks, such as estimating the future position of an object from past observations, are just a few examples of the many regression tasks performed by humans on a daily basis.

The machine learning literature contains many solutions to the regression problem, including nonlinear regression, support vector machines, Gaussian processes and deep neural networks (see [23] for an introduction). It is unclear, however, how humans perform regression tasks. Most of the machine learning solutions rely on the assumption that the mapping from $x$ to $y$ is parametrized by a set of parameters $w$, such that the original regression problem of finding the posterior predictive distribution $p(y|x, D)$ is replaced by a parameter estimation problem, i.e., finding the best set of parameters $w^*$ for the parametrized mapping $p(y|x, w^*)$. However, this approach is not Bayesian since no uncertainty over the parameters $w$ is included in the regression model.

The Bayesian approach to regression proceeds in two steps [24]. First, the posterior distribution over the parameters $p(w|D)$ is computed from the observed data $D$. Then, this posterior is used to compute the posterior predictive distribution by integrating over the parameters:

$$p(y|x, D) = \int p(y|x, w)p(w|D)\mathrm{d}w \tag{1}$$

Taking into account the uncertainty over parameters is particularly relevant for predictions when the size $N$ of the dataset is small compared to the number of parameters. Indeed, taking into account the uncertainty helps to generalize to unknown data and thereby alleviates overfitting.

Parameter uncertainty also plays a key role in computing predictive distribution Eq (1), as estimated, e.g., by the variance of the predictive distribution. In Bayesian decision theory, the predictive distribution is used to minimize the expected cost with respect to the predicted variable. This is important when rewards are unequally distributed, as is the case in many behavioural tasks [25–27]. Some recent work supports the notion that humans make simple decisions in a way which conforms to Bayesian decision theory [12, 28]. In more complex tasks, it has been shown that humans respond suboptimally, which can be largely attributed to noisy inference rather than noisy decision making [29]. A competing decision model to that of Bayesian decision theory is probability matching, wherein random samples are drawn from the predictive distribution. Several studies support the idea that humans use probability matching in cognitive [30, 31] and perceptual tasks [32]. Despite the differences in how prediction uncertainty is used in Bayesian decision theory and probability matching, uncertainty is

nevertheless an integral part of the decision making process in both cases. Both of the afore-mentioned potential pitfalls of the regression problem—overfitting to small datasets and lack of prediction uncertainty—currently limit the power of deep neural network models [33, 34]. These models have millions of parameters and their performance improves with the number of layers [35, 36]. To prevent overfitting, training requires ever larger and more expensive training sets.

It is interesting to note that classic Deep Neuronal Networks (DNNs) do not use weight uncertainty and are therefore limited in their ability to compute prediction uncertainty. Recently, the idea of computing the probability distribution over weights in DNNs and using the distribution for prediction has gained traction and has given rise to the so-called Bayesian Neuronal Network (BNN), for example [37, 38]. Thus, the proposal of BNNs is simply to apply Bayesian regression to DNNs. BNNs promise better performance in the low data regime.

Here, we ask the question whether human observers process parameter uncertainty in accordance with Bayesian regression. We conducted psychophysical experiments in the low data regime with a simple generative model and compared Bayesian regression to other regression models without fitting any hyperparameters other than participant-specific noise. The experimental design made use of the fact that Bayesian regression predicts an uncertainty-modulated transition from a unimodal to a bimodal response distribution. In each trial, we presented participants with 4 points from a hidden, noisy parabola. The task was to correctly extrapolate the parabola, i.e., to find the vertical point of intersection of the parabola with a given horizontal location. The quadratic parameter of each parabola was drawn from a bimodal prior distribution, designed to make the parabolas face either upwards or downwards. After recording the participant's response, we showed the parabola from which the stimulus dots were generated as feedback. This feedback enabled the participants to learn both the prior and the generative model. Because we wanted to test to what extent participants make decisions in accordance with Bayesian regression, we varied the level of noise of the parabola. The rationale is that the higher the noise level, the higher the uncertainty about the correct parameter and, according to Bayesian regression, the more participants should rely on the prior and produce a bimodal response distribution. We found that Bayesian regression indeed explains participants' responses better than maximum likelihood regression and maximum a posteriori regression. Moreover, we compared a loss-based decision model with a sampling-based decision model and found clear evidence for the latter. Indeed, a loss based model with exact inference cannot explain the bimodality of participants' response distributions.

## Results

### A novel paradigm to test regression

We designed a novel psychophysical experiment in which participants had to extrapolate a noisy parabola displayed on a computer screen. In each trial, we chose the parameter $w$ of the parabola $y = wx^2$ from a bimodal prior distribution $\pi(w)$ where the two modes are centered at $w = 1$ and $w = -1$ and the variances are given by $\sigma_\pi^2$ (see Eq 5). The parameter $w$ was either positive (parabola facing upwards) or negative (parabola facing downwards), with the same probability, i.e., 0.5. We selected four dots on the parabola with x-positions close to the parabola's vertex and added zero-mean Gaussian generative noise $\sigma_g$ to the dots' y-positions (see Eq 4). We then presented a fifth dot to the right of the stimulus, always at the same x-position $x^\star = 2$. Participants could move the fifth dot up and down along the y-axis by using the up and down arrow keys. Participants were asked to adjust the y-position so that the dot correctly extrapolated the parabola. During the adjustment task, participants saw only the 4-dot stimulus but not the generating parabola. After the the participant had validated his/her response, we

showed the generating parabola and the adjusted point as feedback. Participants were naive to the purpose of the study. They were not informed about the existence of a prior distribution of the parabola's quadratic parameter, the parabola's bimodality nor the level of generative noise.

In our main experiment, we set the standard deviation of each prior mode to $\sigma_\pi = 0.1$ (if not specified otherwise, assume this value throughout this work) and fixed the values of x-positions to $x_1 = -0.3$, $x_2 = -0.1$, $x_3 = 0.1$ and $x_4 = 0.3$. We generated $j \in (1, \ldots, 20)$ unique stimuli $D_j = \{x_i^{(j)}, y_i^{(j)}\}_{i=1}^4$ at a low (0.03), medium (0.1) and high (0.4) value of the generative noise $\sigma_g$. The rationale is that the higher the noise level, the higher the uncertainty (the lower the likelihood) and the more participants rely on the prior if they act consistently with a Bayesian regression model. At each noise level, we ran 400 trials, repeating each unique stimulus $D_j$ 20 times. The stimulus presentation order was randomized within each noise level (Fig 1B). Thus, we obtained a set of responses for each noise level and for each of the 20 stimuli $\mathcal{R}_j = \{r_1^{(j)}, \ldots r_{20}^{(j)}\}$. The advantage of observing several responses for the same exact stimulus is that we can compare the observed response distribution to the response distributions predicted by the different models.

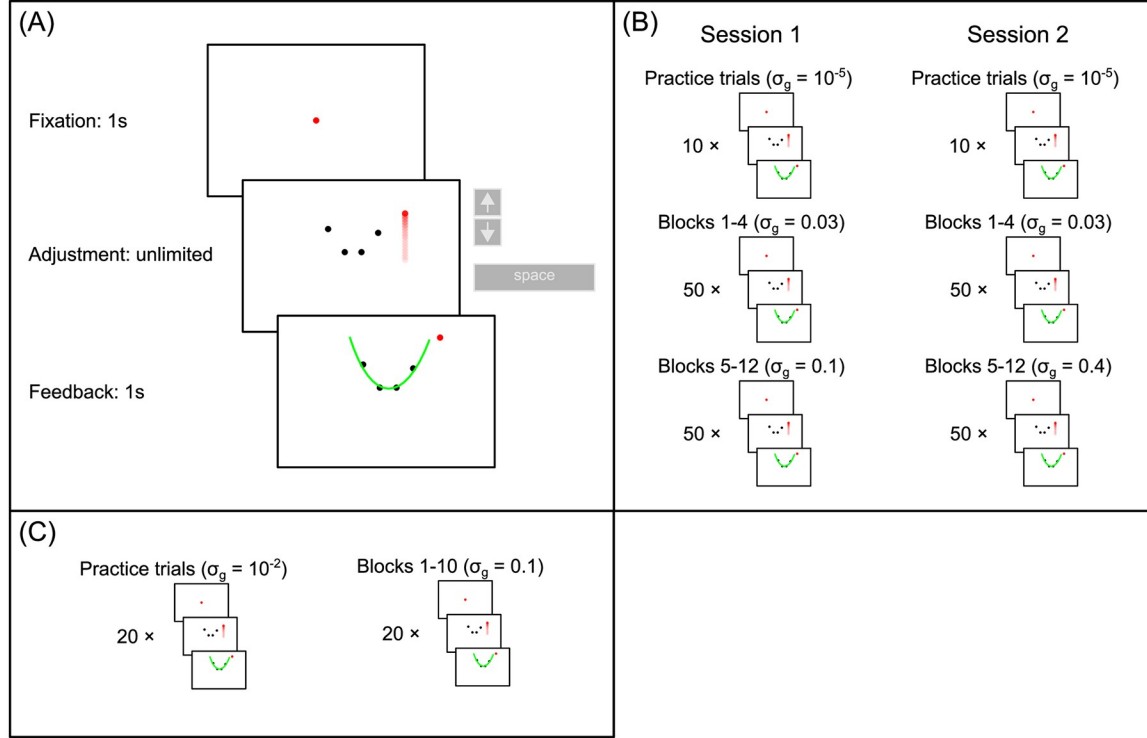

**Fig 1. Experimental protocol.** (A): Procedure of a single trial. First, a fixation dot was presented for 1s before the 4-dot stimulus appeared. Observers then had unlimited time to adjust the fifth dot with the up and down arrow keys. They then clicked the space bar to confirm the final position of the adjustable dot. After the response, the generative parabola was shown for 1s as feedback. (B): Experiment 1: The experiment consisted of two sessions on two separate days. Both sessions began with 10 practice trials with virtually no noise ($\sigma_g = 10^{-5}$), followed by 4 blocks of 50 trials of low noise ($\sigma_g = 0.03$). In session 1, the low noise blocks were followed by 8 blocks of 50 trials of medium noise ($\sigma_g = 0.1$), while in session 2, the low noise blocks were followed by 8 blocks of 50 trials of high noise ($\sigma_g = 0.4$). In total, each participant completed 400 trials per noise level, with 20 repetitions of 20 unique stimuli. In this experiment, $\sigma_\pi$ was set to 0.1. (C): Experiment 2: the experiment consisted of a single session which began with 20 practice trials with very low noise ($\sigma_g = 10^{-2}$), followed by 10 blocks of medium noise ($\sigma_g = 0.1$) trials. Each block consisted of 20 trials, with the generative parabola shown as feedback, as in Experiment 1. Half of the 200 trials consisted of stimuli which were presented just once, while the remaining 100 trials consisted of 10 repetitions of 10 unique stimuli. In this experiment, $\sigma_\pi$ was set to 0.5. See Materials and methods for more details.

Seven naive participants took part in the experiment. We denote the set of all responses of participant $k$ by $\mathcal{R}^{(k)} = \{\mathcal{R}_j\}_{j=1}^{20}$. The stimulus presentation order was identical for each participant. At the beginning of each experimental session, we showed a virtually noiseless 4-dot stimulus $\sigma_g = 10^{-5}$ to familiarise the participants with the task and to estimate their internal noise sources (explained in more detail below).

In a second experiment, we set the prior standard deviation to $\sigma_\pi = 0.5$ and the generative noise to the medium level of $\sigma_g = 0.1$. Instead of using fixed x-positions for the stimuli we added weak Gaussian noise (see Material and methods). We repeated 10 unique stimuli 10 times each, which yielded a total of 100 trials. Four naive participants (different from those recruited for the first experiment) completed the experiment. The key difference to the $\sigma_g = 0.1$ condition in the first experiment was that here, the prior provided much less information about the curvature of the parabolas.

In total, we studied four different conditions: three with $\sigma_\pi = 0.1$ and $\sigma_g \in \{0.03, 0.1, 0.4\}$ in experiment 1, and one with $\sigma_\pi = 0.5$ and $\sigma_g = 0.1$ in experiment 2.

## The regression models

We considered five regression models (see Materials and methods). The Maximum likelihood regression (ML-R) model computes only the point estimate of $w$ that maximizes the likelihood $p(D_j|w)$ and does not make use of the prior distribution at all. The maximum a posteriori model (MAP-R) combines the likelihood with the prior distribution to compute the mode of the posterior distribution $p(w|D_j)$. Despite the fact that it uses the bimodal prior, MAP-R cannot produce bimodal responses because it relies on a point estimate of $w$. The Bayesian regression (B-R) model $\mathcal{M}_{BR}$ takes the entire posterior distribution into account:

$$p(y^\star|x^\star, D_j, \mathcal{M}_{BR}) = \int p(y^\star|x^\star, w)p(w|D_j)dw \tag{2}$$

In B-R, the noise level $\sigma_g$ plays the crucial role of modulating the relative strength of unimodal likelihood and bimodal prior, and hence determines the transition between a unimodal and a bimodal response distribution. Relaxing the assumption that the true generative noise is known, we included an additional variant of B-R that (deterministically) estimates the generative noise $\hat{\sigma}_g$ for the current stimulus $D_j$. We indicate this variant of B-R with a subscript: B-R$_\sigma$. Note that technically, this trial-by-trial noise estimation could also be applied to the other models such as MAP-R. However, MAP-R reduces the posterior distribution over the quadratic parameter to a point estimate. Therefore, an additional trial-by-trial noise estimation would not change its prediction substantially, i.e., it would only shift the unimodal prediction but would not induce a bimodal predictive distribution. Thus, we did not consider a corresponding "MAP-R$_\sigma$" variant. As a null model, we included prior regression (P-R), which replaces the posterior with the prior, i.e., it does not use the likelihood.

For all models considered here the predictive distribution depends deterministically on the 4-dot stimulus. In this sense, they rely on exact inference. Noisy inference is an alternative which assumes that the inference process is corrupted by noise [29]. This alternative would require an additional noise parameter which governs the level of inference noise (see S1 Text). Here, we constrain ourselves to models with exact inference to remain fitting-free, i.e., model predictions for a given stimulus $D_j$ have no free parameter. We used the true values of the hyperparameters because we assume that the participants learned the generative model within a few trials (see S1 Text). Thus, the model predictions require no fitting. For more details, see Materials and methods.

In the plots, we denote the models by the arguments of their predictive distributions, i.e., $y|x, w_{\mathrm{ML}}$ for Maximum Likelihood regression (ML-R); $y|x, w_{\mathrm{MAP}}$ for Maximum a Posteriori regression (MAP-R); $y|x, D$ for Bayesian regression (B-R); $y|x, D, \sigma_g$ for Bayesian regression with noise estimation (B-R$_\sigma$) and $y|x$ for prior regression (P-R).

## The decision models

We considered two decision models that turn the predictive distributions into a response distribution: probability matching and Bayesian decision theory. In the case of probability matching, it is assumed that participants draw random samples from the predictive distribution: $y^\star \sim p(y|x, \mathcal{D}, \mathcal{M})$. If not stated otherwise, we use sampling-based decisions throughout this work. Meanwhile, according to Bayesian decision theory, participants select a response by minimizing the expected loss function $y^\star = \arg\ \min_{y^\star} \langle L(y, y^\star)\rangle_{p(y|x,\mathcal{D},\mathcal{M})}$. Here, we considered only the square loss, which is equivalent to the choosing mean of the predictive distribution $y^\star = \langle y \rangle_{p(y|x,\mathcal{M})}$. Independently of the form of the loss function, the Bayesian decision theory generates responses from the predictive distribution deterministically. When we use loss-based decision models, we indicate this by adding the prefix "$L$": to the model, e.g., $L{:}\, y|x, D$ for a loss-based decision model applied to Bayesian regression.

To model participants' responses, we also accounted for internal sources of noise, i.e., noise which is inhere to neural processing, decision making and the execution of motor action [29, 39]. We call the sum of these noise components motor noise for brevity. The motor noise is not a model parameter but a participant-specific parameter. We computed the motor noise $\sigma_m$ for each participant from the 20 responses to the noiseless stimulus. To ensure robustness to outliers, we used the average value between the 16% and 84%-percentile of the response distribution. The values of motor noise for the seven participants of the main experiment were $\sigma_m^{(1,\ldots,7)} = (0.22, 0.3, 0.74, 0.88, 0.34, 0.37, 0.54)$, respectively while for the second experiment, we used the average of these values, i.e., 0.48 because the noise-free responses of experiment 2 were not available. The motor noise was included in the models by convolving the predictive distribution with a Gaussian of variance $\sigma_m^2$. In the case of loss-based decision models, motor noise was the only source of response variability.

## Modality of predicted and observed response distributions

Fig 2(A) shows the responses of a representative participant along with the predicted response distributions of the different models. Both ML-R and MAP-R ignore one of the modes (here, the mode corresponding to a downward-facing parabola). In addition, the parabola predicted by ML-R has lower curvature than the parabolas predicted by any of the other models (i.e., the absolute value of the ML-R parabola's quadratic parameter is lower) than that of the parabola that the participant responded with. A potential explanation for this finding is that, while ML-R does not take the prior into consideration, humans do make use of the prior. In the low noise regime ($\sigma_g = 0.03$, Fig 2(B)), the discrepancy between the participant's response distribution and the prior regression model's predicted response distribution in terms of the number of modes (unimodal and bimodal, respectively) rules out the explanatory validity of the latter. In the higher noise regimes ($\sigma_g \in (0.1, 0.4)$, Fig 2(C) and 2(D)), MAP-R and ML-R fail to account for the fact that the participant's responses are distributed across both modes. The finding that at $\sigma_g = 0.03$ MAP-R matches the participant's responses often with high accuracy provided implicit evidence that participants used the prior and had learned the parabola's generative model.

Fig 2(E) illustrates the participant's responses in the condition when $\sigma_\pi = 0.5$. The participant's responses cover a wider range of values than in the conditions of experiment 1 when $\sigma_\pi$

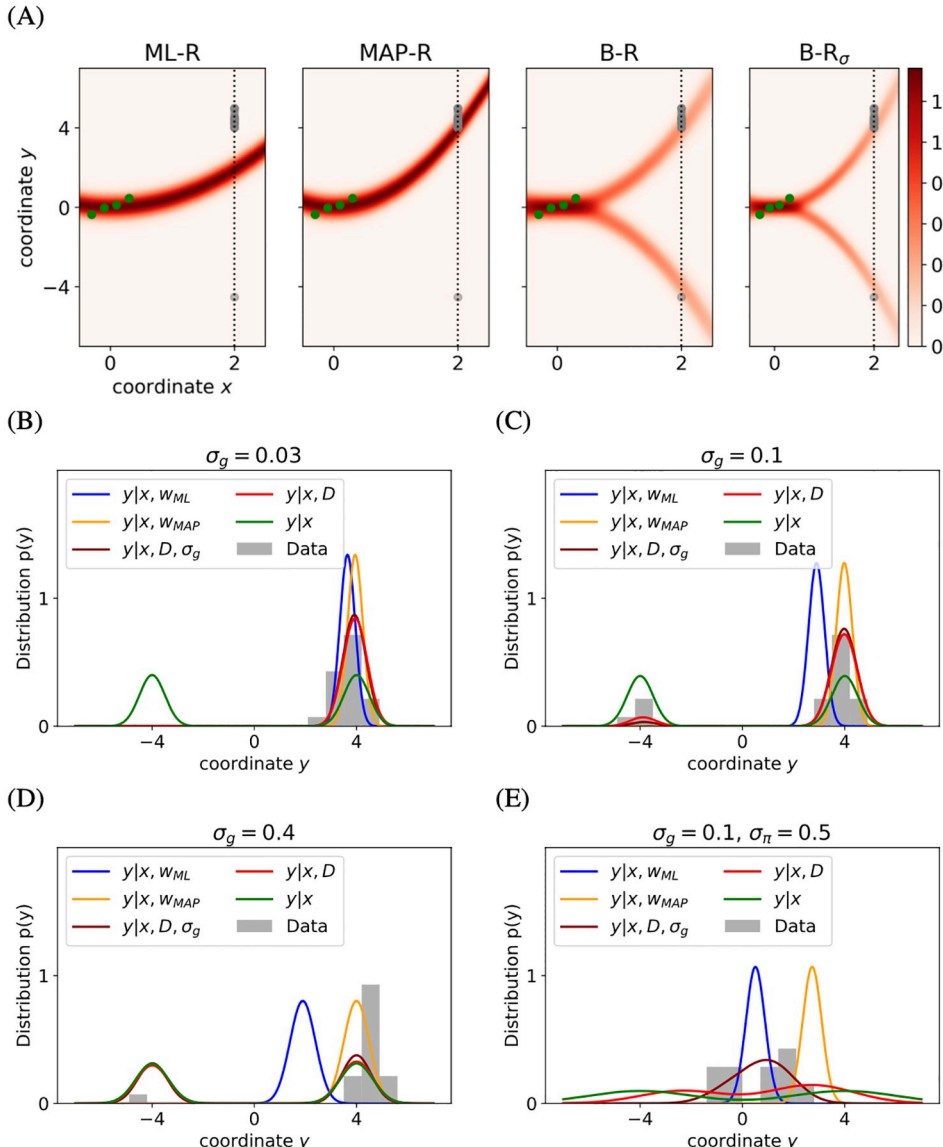

**Fig 2. Example responses.** B-R is the only model that can explain the transition from unimodal response (at low noise, (B)) to bimodal response distribution (at high noise, (D)). (A) A sample stimulus (green dots) at high noise level ($\sigma_g$ = 0.4). For this specific stimulus, contours indicate the response distributions predicted by ML-R, MAP-R, B-R and B-R$_\sigma$ (not shown to the participant) at various $x^\star$. At $x^\star$ = 2, we recorded the participant's responses (gray dots). The cross section at $x^\star$ = 2 is shown in (D). (B—E) The predicted response distributions at $x^\star$ = 2 of ML-R (blue), MAP-R (orange), B-R (red), B-R$_\sigma$ (dark red), P-R (green) and observed responses (gray). As $\sigma_g$ increases (B—D), the data becomes less informative. Consequently, and in accordance with B-R, the response distribution becomes more bimodal. (E) Due to the weak prior the predictions of B-R and B-R$_\sigma$ respond more strongly to the data and diverge from the modes of P-R more stronlgy than in the previous conditions. The skewness of B-R$_\sigma$ results from the mixture of both Gaussian components.

is smaller (i.e. $\sigma_\pi$ = 0.1). While the generative noise $\sigma_g$ = 0.1 is the same as in Fig 2(B), this condition is more difficult because the prior is less reliable. As a consequence participants rely more strongly on the noisy stimulus and produce more response variability. In this example, the responses are closely clustered around the center. B-R$_\sigma$ is attracted more strongly to the

center than B-R because the former is more driven by the stimulus due to underestimating the noise.

## B-R outperforms the other models

Fig 2. In order to formally assess model performance, we next conducted a quantitative model comparison across all participants. For each of the seven participants individually, we computed the log probability that the participant's responses arise from the given model. We summed these log probabilities for all of the unique stimuli $D_j$ as a measure of the quality of the model. Fig 3(A) shows these values relative to the B-R baseline value for each noise level, averaged across participants. Negative values indicate poor performance relative to B-R. A subject-level analysis showed that the model comparison results were not driven by any single participant's data (see S1 Text). Because our model comparison is fitting-free, we do not need to account for different levels of model complexity. Indeed, in the present case, the log likelihood comparison is equivalent to using the Bayesian Information Criterion.

As Fig 3(A) shows, B-R is among the highest performing models for all conditions. As the task difficulty increases (left to right), P-R performance approaches that of B-R. This is because the parameter uncertainty encoded in the prior becomes more important and the response distribution becomes bimodal. Neither MAP-R nor ML-R can capture this and therefore perform

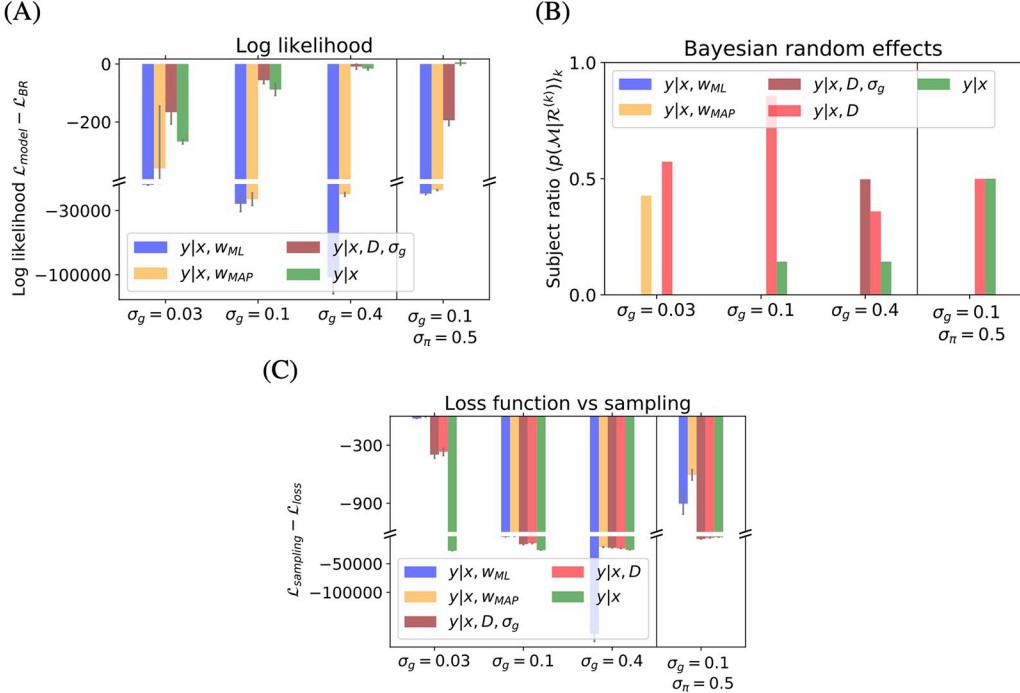

**Fig 3. Model comparison.** The model comparison shows that the B-R model best explains the data (A, B) and that sampling-based decision models outperform loss-based decision models (C). (A) Difference in log likelihood with respect to B-R averaged over participants for different experimental conditions. Negative values mean that B-R wins the comparison. B-R is either winning ($\sigma_g \in (0.03, 0.1)$) or equivalent to P-R because the two coincide at high levels of parameter uncertainty ($\sigma_g = 0.4$ and $\sigma_\pi = 0.5$). (B) The expected likelihood of each model for a randomly selected participant shows what fraction of participants are best described by a model. Overall, B-R and B-R$_\sigma$ describe the population best. (C) Log likelihood difference between a sampling and a loss-based decision model. Negative values favour sampling. At all other conditions and for all regression models, sampling explains the data better than loss-based decision models with exact inference. For B-R, B-R$_\sigma$ and P-R, loss-based models do not predict bimodal responses. At low noise $\sigma_g = 0.03$, loss-based models underestimate the response variance. Error bars represent the SEM across participants.

poorly. These results are consistent across participants (see S1 Text for a subject-level analysis).

At low noise $\sigma_g = 0.03$, participants give unimodal answers and the mean predictions of B-R and MAP-R are indistinguishable. Then the model that better captures the response variability wins. In general, B-R explains the variability of the responses better. The variability is also the reason why P-R performs relatively well under the more difficult conditions, i.e., $\sigma_g = 0.03$ and $\sigma_\pi = 0.5$.

The averaged results are largely consistent with a subject-level analysis. A notable exception is that at $\sigma_g = 0.03$, MAP-R emerges as the best model (closely followed by B-R) for participants 3, 4 and 7 (see S1 Text). A Bayesian random effects analysis confirms this. Specifically, we used the model posterior $p(\mathcal{M}|\mathcal{R}^{(k)})$ averaged over a randomly selected participant $k$. This measure reflects the ratio of participants for which model $\mathcal{M}$ wins. Fig 3(B) shows that the responses of the majority of participants are best modelled by B-R or B-R$_\sigma$. Since B-R interpolates between MAP-R (at low noise) and P-R (at high noise), as expected, at $\sigma_g = 0.03$, i.e., the easiest condition, the responses of some participants are also well modelled by MAP-Rwhile at the most difficult condition, i.e., when $\sigma_\pi = 0.5$, the responses of half of the participants are best described by P-R (and the other half by B-R).

Next, we investigated if sampling or the loss function perspective explains the responses better. Fig 3(C) depicts the log likelihood of loss-based decision making compared to sampling for each model. Negative values indicate that sampling wins. Sampling explains the data better for all models and in all experimental conditions. One explanation is that the loss mechanism turns bimodal predictive distributions into unimodal predictive distributions. Here, we use the square loss such that the (unimodal) response distribution is centered on the mean of the predictive distribution. In the case of P-R, the mean of the predictive distribution lies at the center of both modes. Clearly, this method does not capture bimodal responses. This is why the performance difference between the two decision models is smallest at $\sigma_g = 0.03$, where all models except for P-R make predictions which are close to unimodal.

The second explanation for the better performance of sampling is that the loss function approach with exact inference underestimates response variability. The response variability differs from one stimulus to another and is often higher than $\sigma_m$. This explains the better performance of sampling for MAP-R and ML-R, since the effect of turning a bimodal response distribution into a unimodal one is absent. In these cases, the sampling-based decision model has the effect of increasing the variance of the predicted response distribution by $\sigma_g^2$. This leads to better model performance on variable response data, even in the experimental conditions $\sigma_g = 0.03$ where participants respond unimodally.

In conclusion, from the models considered here, B-R with sampling best explains participants' responses.

## B-R explains the generative noise-dependent increase in response variance

A key characteristic of B-R is the transition of the model's posterior predictive distribution from unimodality to bimodality as $\sigma_g$ increases, i.e., as the data become less informative. To analyse this transition, we used the participants' response variances at the different levels of generative noise.

The variance of the response distributions is sensitive to bimodality. For example, if all responses are distributed evenly across both modes, the variance is close to 16, which corresponds to the variance of the prior P-R. If all responses are located in a single mode, the variance is typically smaller by a factor of ten (e.g., see Fig 2(B)). We explain this in more detail below.

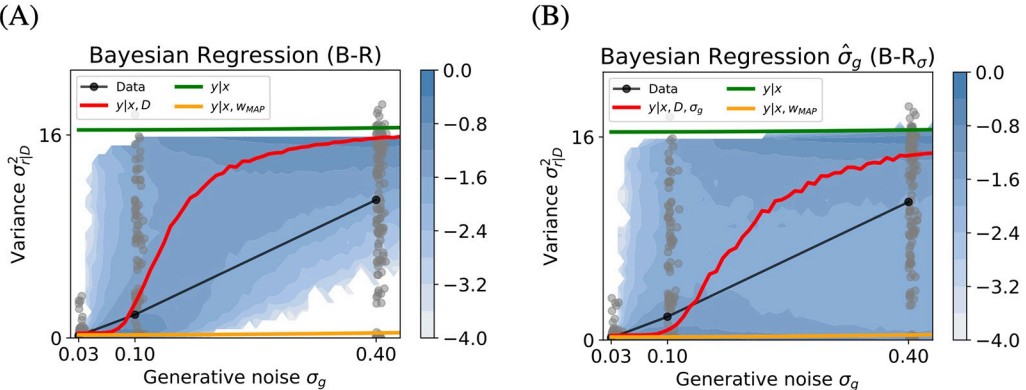

**Fig 4. Response variances of predicted and empirical distributions, as a function of generative noise.** B-R best explains the increase in response variance as a function of the generative noise $\sigma_g$. Variances of the empirical response distributions from all participants (gray dots, median: gray line) and predicted response distributions, corresponding to the two B-R variants (median: red line, log probability: heatmap). B-R (A) Interpolating between MAP-R and P-R, only the B-R variants capture the upward trend in the data. At $\sigma_g = 0.4$, B-R fails to account for the empirical responses with close-to-zero variances. (B) At $\sigma_g = 0.4$, B-R$_\sigma$ predicts a bimodal variance distribution because, in trials with low noise estimates, the predicted response distribution is unimodal and thus variance is low. Because of these low-variance trials, the median of B-R$_\sigma$ increases slower than the median of B-R and captures the empirical median better. Because ML-R and MAP-R behaved identically, the MAP-R represents both regression models.

For each stimulus $D_j$ and for each participant, we computed the variance of the 20 observed responses $\sigma^2_{r|D_j}$. Because we have 7 participants, this yields a distribution over $7 \times 20 = 140$ empirical variance values at each value of $\sigma_g$. We compared this distribution with the response variance distribution predicted by the models. To achieve a higher resolution and show the dynamics of the variance as a function of the generative noise, we generated 5000 unique stimuli from a densely spaced $\sigma_g$ instead of relying on the small number of stimuli and noise levels used in the experiment (see Materials and methods).

The empirical variance distribution (gray) and its median (black) are shown in Fig 4 along with the predicted median of the variance distribution for each model (color coded). For B-R and B-R$_\sigma$, we plotted the distribution in Fig 4(A) and 4(B), respectively.

The median of the B-R variance distribution increases with the noise level. This is due to the fact that B-R's predicted response distribution transitions from unimodal to bimodal; this transition is modulated by the generative noise, which determines the relative contribution of the prior and likelihood to the response distribution. Consequently, the B-R model is the only one for which the variance values smoothly transition from the MAP variance at the low noise level ($\sigma_g = 0.03$) to the P-R variance at the high noise level ($\sigma_g = 0.4$). The P-R variance remains constant and the MAP-R variance increases very weakly as a function of the generative noise. The variance analysis provides further evidence for the superiority of both B-R variants over the other models.

While B-R captures the general trend in the data, it fails to account for two key characteristics. First, the median variance increases slower than B-R would predict, and secondly, at high levels of generative noise, B-R fails to reproduce the lower part of the distribution (where response variances are close to zero). A potential explanation for this discrepancy is that participants estimate the noise on a trial-by-trial basis. When the noise added to the 4-dot stimulus was, by chance, such that the dots appeared to be well-aligned on a parabola, participants would, presumably, underestimate the generative noise and respond in a way which was consistent with a unimodal distribution. The fact that the B-R$_\sigma$ model captures the empirical

variance better than B-R provides some evidence for this idea. On some trials, B-R$_\sigma$ underestimates the true $\sigma_g$ and applies the B-R formalism with high confidence in the stimulus data. In these cases, the model relies strongly on the likelihood and bimodality, which normally enters through the prior, is not achieved. Rather, the resulting response distribution is unimodal and has low variance.

Despite the fact that B-R$_\sigma$ describes the qualitative features of the variance distribution better than B-R, it performs worse in terms of log likelihood. This shows that the low variance responses of humans and of B-R$_\sigma$ do not always coincide on a trial-by-trial basis.

To better understand the relation between response variance and bimodality, we dissect the variance of a bimodal response distribution to stimulus $D_j$ into its components:

$$\sigma^2_{r|D_j} = \sigma^2_m + \sigma^2_{y|D_j} + c(1-c)(\mu_1 - \mu_2)^2, \tag{3}$$

where $\mu_1$ and $\mu_2$ are the means of the modes of the posterior predictive distribution, $c$ is the mixture coefficient and $\sigma^2_{y|\mathcal{D}_|} + \sigma^2_m$ corresponds to the variance of both modes. The unimodal contribution is not mode-specific because we chose a symmetrical prior. The first two terms constitute a unimodal contribution and the last term a bimodal contribution. The latter is controlled by the mean dispersion $(\mu_1 - \mu_2)^2$ and a prefactor $c(1-c)$ that is equal to zero for $c \in \{0, 1\}$ and is maximal for $c = 1/2$. To determine to what extent each component of this dissection is present in the response data, we defined the empirical counterparts of $\mu_1$, $\mu_2$ as the means of the upper and lower modes of the response distribution and $c$ as the mixture coefficient, corresponding to the fraction of positive responses $r > 0$. For the unimodal variance contribution $\sigma^2_{y|D_j} + \sigma^2_m$, we used the variance of the mode which contains the majority of responses (see Methods for more details). The comparison between data and models shows that both B-R variants correctly predict the driver of the observed variance to be the transition to bimodality. Fig 5(A) shows the predicted positive coefficient $c$ (median) of the models as a function of the empirically-observed coefficient across all participants and stimuli (in the main experiment). As further evidence for the validity of the B-R perspective, both B-R variants correctly predict the fraction of positive responses. Indeed, the smooth transition from a unimodal to a bimodal distribution is nicely captured by B-R. In contrast, MAP-R transitions sharply and is more reminiscent of a step function while P-R predicts equally strong modes across all noise level conditions.

The bimodal distribution of responses depends on the prefactor $c(1-c)$ and the mean dispersion. Fig 5(B) shows the median value of the prefactor as a function of generative noise (across all participants and stimuli). Data and model predictions qualitatively match the behaviour of the variance in Fig 4. Indeed, the other contributions to the variance are less important. The mean dispersion, shown in Fig 4(C), plays the role of a large constant. The unimodal contribution to the variance, shown in Fig 5(D), is small compared to the bimodal contribution. In conclusion, the coefficient $c$ plays the dominant role in determining the variance of the response distribution. Because the two B-R variants estimate $c$ sufficiently well, they best match the empirical variance distribution.

Interestingly, all models overestimate the unimodal variance, with the exception of MAP-R in the low noise condition (Fig 5C). The B-R variants predict larger variance than MAP-R because they translate the posterior parameter into response uncertainty. P-R predicts even larger response variance because it uses the prior parameter uncertainty which is generally larger than the posterior one. Despite the fact that MAP-R best describes the median variance, it performs worse than B-R in terms of log likelihood. Fig 5(C) reveals that one factor contributing to the poorer performance of MAP-R is the occurrence of unimodal, high variance responses.

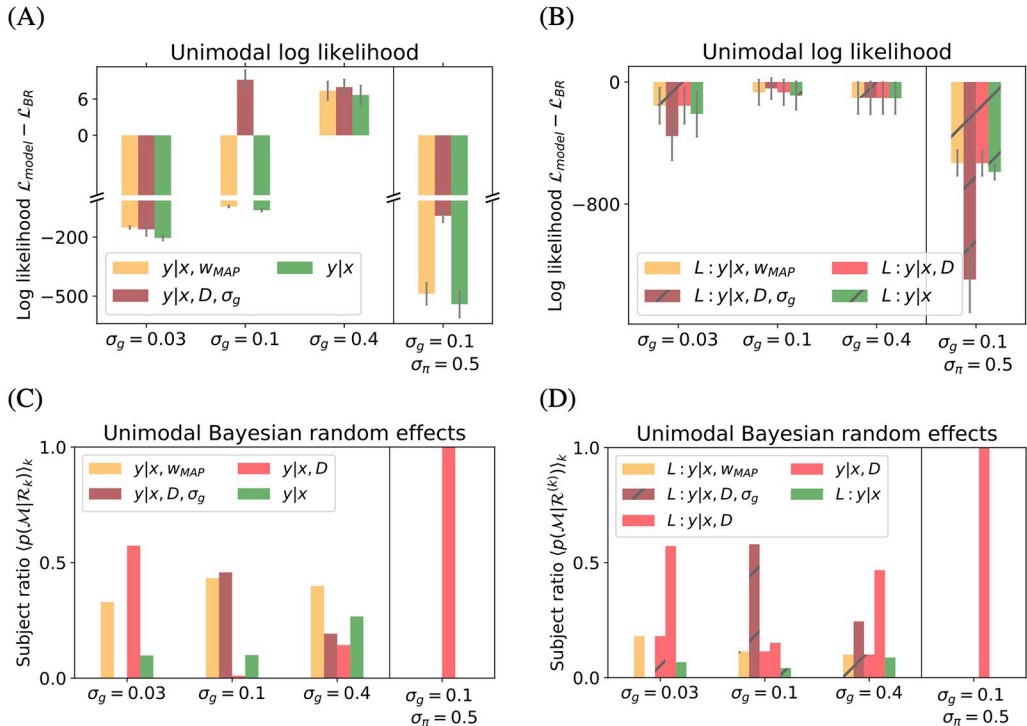

**Fig 5. Median of each of the bimodal response distribution variance components across all participants and stimuli.**
(A) Predicted coefficient of positive mode as a function of the empirical coefficient (across all noise levels). ML-R behaves identically to MAP-R. Thus, the MAP-R curve represents both models. The shaded area shows the 40% and 60% quantiles. (B) Prefactor of bimodal contribution as a function of generative noise. Data jittered for visibility. (C) Unimodal contribution to the variance. Empirical variance computed on mode with majority of responses. (D) Mean dispersion. Only trials with bimodal responses included. As the stimulus becomes more noisy, human responses and B-R variants conform to the prior.

In summary, the variance analysis provides further evidence that B-R captures the way in which generative noise induces a transition from unimodality to bimodality in participant responses. However, B-R overestimates response variance. Trial-by-trial estimation of the noise offers a potential explanation for why participants cluster their responses more unimodally than predicted.

## Unimodal responses are overall best explained by B-R

Thus far, the main factor behind the superiority of the B-R model's performance relative to the other models is the ability of B-R to capture the bimodality of responses, i.e., to correctly set the mixture coefficient. However, the previous analysis showed that unimodal variance decreases as a function of generative noise while B-R predicts an increase. It remains unclear if B-R still wins the model comparison in a unimodal setting where performance is independent of the mixture coefficient.

To address this question, we conducted a model comparison on a unimodally conditioned dataset. For each stimulus $D_j$, we considered only responses in which the dominant response mode and the dominant mode of the model predictions coincided (see Methods for details). The conditioning yields a unimodal dataset in the sense that all predictions and responses belong to the same mode. To make the model comparison fair for the bimodal predictive

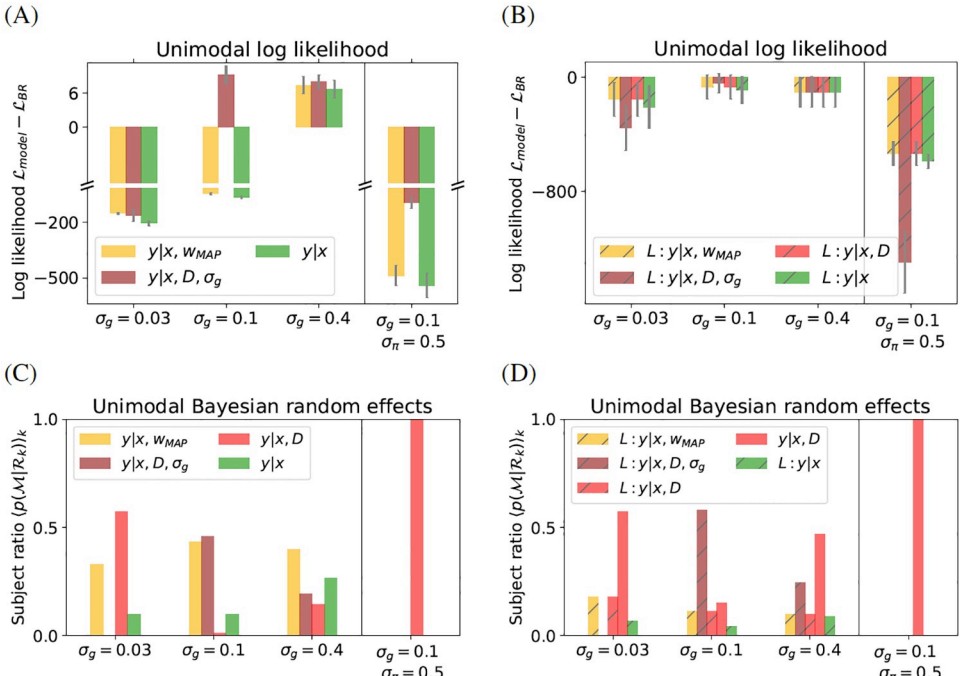

**Fig 6. Unimodal model comparison.** The unimodal analysis confirms previous results: overall B-R with sampling wins the model comparison. (A) Differences in log likelihood on unimodal data, averaged over participants. Negative values mean that B-R wins. ML-R is omitted because its poor performance complicates visualisation. (A) B-R wins at $\sigma_g = 0.03$ and $\sigma_\pi = 0.5$, but not in the other conditions. (B) All models use the quadratic loss function to select responses, with response variance given by the motor noise $\sigma_m^2$. B-R with sampling explains the unimodal data best for most participants. High subject-level variability results in large errors (see S1 Text for a subject-level analysis). (C, D) The fraction of participants best described by a given model. At $\sigma_g = 0.4$, several models perform well. Error bars indicate SEM across participants.

distributions of P-R and the B-R variants, we removed the inferior mode and normalised the remaining probability mass to one.

At the group level, B-R wins the model comparison across all conditions, as shown in Fig 6 (A). B-R clearly outperforms the other models in two conditions in particular: at $\sigma_g = 0.03$ and at $\sigma_\pi = 0.5$.

Interestingly, in the bimodal dataset, B-R did not emerge as a clear winner at $\sigma_\pi = 0.5$ because P-R performed similarly well. Thus, B-R is not better than P-R at modelling how participants balance the two modes, but once the mode is chosen, it performs better. At $\sigma_g = 0.1$, B-R and B-R$_\sigma$ outperform the other models. B-R$_\sigma$ wins by a small margin (see axis scaling). However, a subject-level analysis (see S1 Text) shows that the average is mostly driven by participant 1, while in the case of other participants all models perform similarly well. At $\sigma_g = 0.4$, no clear winner emerges. Intuitively, this makes sense because the stimulus is not informative and all models rely mostly on the prior information about $w$. Here, B-R wins or performs similarly to other models.

The Bayesian random effects analysis (results shown Fig 6(C)) confirms the previous results. The ratio of participants whose responses are best described by a given model $\langle p(\mathcal{R}^{(}k)|\mathcal{M})\rangle_k$ indicates that B-R describes the population at $\sigma_g = 0.03$ and at $\sigma_\pi = 0.5$ well. As in the bimodal dataset, MAP-R reflects the responses of some participants well at $\sigma_g = 0.3$. At

$\sigma_g = 0.1$, the log likelihood performance (Fig 6A) of all models is similar but B-R$_\sigma$ and MAP-R win by a small margin (see S1 Text for a subject-level analysis). Hence, B-R$_\sigma$ and MAP-R perform best in the Bayesian random effects analysis (Fig 6C). No clear winner emerges at $\sigma_g = 0.4$.

Next, we revisit the question of whether participants sample or use a loss function. In the bimodal data, the loss function approach was at a disadvantage because it could only produce unimodal response distributions. This disadvantage is not present in the unimodal dataset. To make the performance of models in Fig 6(A) and 6(B) comparable, we use B-R with sampling as the baseline in both plots. Fig 6(B) shows that, averaging across participants and conditions, B-R with sampling outperforms the loss-based models. The large errors in (B) reflect large intersubject variability. The Bayesian random effects analysis in Fig 6(D) confirms that B-R also wins at the subject-level at $\sigma_g = 0.03$ and at $\sigma_\pi = 0.5$. One exception is L:B-R$_\sigma$ at $\sigma_g = 0.1$. Indeed, the subject-level analysis (S1 Text) shows that in terms of the averaged log likelihood at middle and high noise $\sigma_g \in (0.1, 0.4)$ participant 1 is an outlier. For other participants, the performance of B-R with sampling and the loss-based models is very similar. Despite the higher intersubject variability in the case of the unimodal dataset than in the bimodal dataset, the unimodal analysis provides convincing evidence of the superiority of B-R with sampling over other models considered here. In contrast to the model comparison in the bimodal analysis, in the unimodal case B-R clearly wins the model comparison at $\sigma_\pi = 0.5$.

## Discussion

In our experiment, participants adjusted a dot such that it coincided with on a parabola determined by four other dots. We used the log likelihood to compare participants' responses to the predictions of ten models: five regression models ML-R, MAP-R, P-R, B-R and B-R$_\sigma$ combined with two decision models, i.e., probability matching (sampling) and Bayesian decision theory (loss-based). B-R with sampling best explained the responses across various experimental conditions. An analysis of the observed and predicted response variance showed that the model comparison results were mainly driven by the transition from unimodal to bimodal responses. Only the B-R variants were able to capture this aspect of the data. However, participants clustered their responses more often in one of the mode than B-R predicted. This resulted in a discrepancy between the predicted and empirical response variances. For B-R$_\sigma$, this discrepancy was smaller. Thus, one possible explanation for the discrepancy is that participants were estimating noise on a trial-by-trial basis. Since in the variance analysis we considered the response variance from all trials, the relatively better performance of B-R$_\sigma$ here did not translate into superior performance in the log likelihood analysis, in which we analyzed responses on a trial-by-trial basis. B-R without noise estimation was more accurate in predicting the mean and variance of the data trial-by-trial. In a final analysis, we conditioned the responses to a single mode to eliminate the effects of bimodality which was the driving factor behind model comparison results in the first two analyses. This allowed us to study the performance of B-R based on its mean and variance. The analysis of the unimodal dataset generally confirmed the previous results. B-R$_\sigma$ with sampling either outperformed or performed similarly to the other models.

Our results suggest that humans turn the posterior predictive distribution into a response via probability matching rather than Bayesian decision theory. The loss function approach fails to explain the bimodality of responses to repeated identical stimuli. One way to interpolate between Bayesian decision theory and probability matching is to present distributions by samples [40]. The number of samples used to approximate the (predictive) distribution interpolates between both decision models. If the number of samples is sufficiently large, the

approximated distribution converges to the true distribution, and we enter the domain of standard Bayesian decision theory. However, if only a single sample is used for the approximation, probability matching is recovered. This is because applying Bayesian decision theory to a one-sample distribution returns the location of this sample as a response. The number of samples takes the role of a transition parameter between classical Bayesian decision theory and probability matching.

[29] In the context of a categorical decision task, Drugowitsch et al. [29] showed that noisy inference (rather than noisy decision making or noisy perception) explains the largest fraction of participants' response variability. Indeed, noisy inference offers an interesting way to reconcile Bayesian decision theory with bimodal responses. Conceptualising the choice between two modes as noisy inference over two unimodal models leads to a bimodal response distribution (see S1 Text). At low generative noise, the noisy inference procedure yields a unimodal response distribution because the difference between the two model evidences is large. At high generative noise, the evidences for both models are similar such that the inference noise becomes the decisive factor in the participant's response. In this case, noisy inference predicts a bimodal response distribution. As in Bayesian regression, the transition from unimodal to bimodal response distribution depends on generative noise. However, the speed of the transition also depends on the inference noise, i.e., a free parameter. In Bayesian regression, this transition speed is computed as a function of the stimulus and the parameters of the generative model, and no fitting is required. Because we wanted to study how humans process parameter uncertainty in a fitting-free context, we did not test the noisy inference model quantitatively. Future work is required to further explore the relationship between Bayesian regression and noisy inference.

Throughout this study, we assumed that the generative model is known. In real world regression tasks, this assumption is typically not justified. Instead, subjects must simultaneously learn the generative model and its parameters. For example, in the context of our experiment this would translate to not informing participants ahead of time that the 4-dot stimuli were generated from parabolas. Bayesian regression extends naturally to tasks with model uncertainty. The Bayesian approach to making predictions makes use not only of the expectation over the posterior over the parameters but also of the expectation over the posterior over the models. Thus, Bayesian regression with model uncertainty requires subjects to infer the posterior over models and to average over this posterior. Compared to Bayesian regression with a known generative model, this multiplies the computational burden by the number of relevant models. It is an interesting question whether subjects solve regression tasks with model uncertainty by taking advantage of Bayesian regression or whether they rely on point estimates such as the MAP-estimator of the model posterior. A study in the context of sensory fusion suggests the latter [41] but it is unclear to what extent this is also the case in the domain of function fitting.

Compared to toy examples, real world tasks typically involve complex models with high-dimensional parameter spaces. This makes the evaluation of the integral in Bayesian regression particularly difficult. Sampling offers a potential solution because it scales well to high dimensions and integrals reduce to the evaluation of a sum. Recent advances in neuronal algorithms [42, 43] suggest that, in theory, the brain can efficiently encode probability distribution via samples. Thus, sampling provides an intriguing direction to further explore potential links between psychophysical experiments and neuronal implementation of uncertainty.

For a given generative model, Bayesian regression and other regression models prescribe how to make prediction when parameter uncertainty is present. For example, MAP-R uses a point estimate of the posterior while B-R uses the entire posterior. Thus, the performance of a regression model in terms of its ability to model human responses depends on two factors: the

ability of the regression model to describe how humans handle uncertainty and the degree to which the theoretically-chosen generative model is true to the generative model inferred by the observer. The predictions of the regression models in our study are limited in that they assume a parabolic generative model. A previous study reiterated the formal equivalence of Bayesian regression and Gaussian processes and demonstrated the flexibility with which Gaussian processes can model human responses in a complex function fitting task [19]. In the case of the Bayesian regression model, the authors fit various hyperparameters, and it was unclear how they controlled for the complexity of the fit. Thus, the study could not answer if the Bayesian regression model performed well because of its flexibility in representing different generative models or because it captured how humans process parameter uncertainty. In our work, we removed the confounding factor by enforcing a simple generative model through feedback in every trial. Instead we remained fitting-free and could, thus, study directly how participants processed parameter uncertainty. The rationale of simplicity rather than complexity has advantages for the analysis as well. The different models are analytically tractable and thus can be studied systematically. Additionally, the one-dimensional response space was easy to visualize and the amount of data needed to compare the predictive and empirical distributions was limited.

To remain fitting-free, we assumed that participants know the generative model, including the prior over the parameters. Without this assumption, we would have had to account for potential temporal dynamics of learning with a participant-specific, time-dependent prior. For instance, it might take participants a non-negligible amount of time to learn the generative model or their responses could be influenced by immediately preceding trials. To avoid such complications, we showed the generative parabola after each trial and we chose a function that humans can learn [21], i.e., a parabola. Indeed, after having run the experiments, we found that there was no substantial learning taking place between the first and the last trials except for some mild learning at $\sigma_g = 0.1$ (see S1 Text). To extend our study to continuous learning, it would be interesting to relax the i.i.d. assumption of the stimuli in the generative model, as in [44], and investigate if a Bayesian framework models the evolution of posterior parameter uncertainty as well.

We presented and analysed our experimental task within the framework of regression. After seeing the training data, i.e., the 4-dot stimuli, participants were asked to make predictions. Then, one way of making predictions is to compute the posterior predictive distribution by marginalising over the posterior of the model parameters. Alternatively, the task can be interpreted as inference of the point where the parabola intersects a vertical line at a chosen x-position given the 4-dot stimulus. The posterior predictive corresponds to the posterior of the response location given the data. Indeed, there is a formal equivalence between Bayesian regression with linear Gaussian generative models and Gaussian processes with a kernel that encodes the generative model (e.g. [19]). Algorithmically, however, Bayesian regression and Gaussian process inference differ. B-R focuses on the compression of training data into model parameters or a distribution of model parameters, e.g., the MAP estimator or posterior. The training data does not need to be stored to make new predictions. In contrast, Gaussian process inference requires that the training data is stored. Thus, the memory requirements grow linearly with the size of the training data, which constitutes an important drawback of Gaussian process models. To distinguish between the B-R and Gaussian process inference perspectives, one would need to design regression tasks that cannot be reformulated as inference because observers can neither see nor remember the entire stimulus when they make predictions. One could achieve this by sequentially presenting many training data such that memorization is not a viable option but sequential updates of the posterior are.

Our work was inspired by the growing emphasis on parameter uncertainty in the machine learning community; however, it is important to highlight that function learning and extrapolation have been studied before. The function learning literature has addressed which types of functions humans can learn [45], how batch or sequential data representation affects learning [22], to what extent human behaviour can be modelled by parametric functions [46] and how well humans extrapolate [21]. However, to the best of our knowledge, these studies have so far failed to conduct a minimal experiment to establish that humans process parameter uncertainty in accordance with Bayesian regression. Our contribution will help to better understand the brain's remarkable ability to learn and generalise from very little data and underpins the power of Bayesian regression as a framework in psychophysical modelling.

## Methods and materials

### Stimulus generation from the bimodal prior

Here, we describe in detail how stimuli are generated. On the $j^{th}$ trial, participants are presented with a stimulus consisting of $N = 4$ points in a 2-dimensional space:

$D_j = \{(x_i^{(j)}, y_i^{(j)})\}_{i=1}^N$. For the main experiment (with $\sigma_\pi = 0.1$, see below), we fixed the x-values to $(-0.3, -0.1, 0.1, 0.3)$ respectively. For the additional experiment (with $\sigma_\pi = 0.5$, see below), we drew the x-values from Gaussians with means $(-0.18, -0.09, 0, 0.09)$ and standard deviation 0.09 but resampled if the minimal distance was less than 0.1 between any two points. In both cases, we then generated the y-coordinates from a Gaussian generative model with a parabolic non-linearity and the *generative parameter*, $w_j$:

$$p(y|x, w_j) = \mathcal{N}(y; w_j x^2, \sigma_g^2) \tag{4}$$

The parameter $w_j$ is drawn from a mixed Gaussian prior

$$\pi_\gamma(w_j) = (c\mathcal{N}(w_j; \mu_\pi, \sigma_\pi^2) + (1-c)\mathcal{N}(w_j; -\mu_\pi, \sigma_\pi^2)) \tag{5}$$

where the parameter set $\gamma = (\mu_\pi, \sigma_\pi^2, c)$ consists of the mean $\mu_\pi = 1$, mixing coefficient $c = 1/2$ and the standard deviation $\sigma_\pi = 0.1$ for the main experiment and $\sigma_\pi = 0.5$ for the additional experiment. We denote the total set of hyperparameters (suppressed for notational clarity), from the prior and the generative probability, by $\alpha = (\gamma, \sigma_g^2)$. Each parameter $w_j$ corresponds to a *generative parabola*. Given this model and given a stimulus $D_j$, we asked participants to predict the y-component $y^\star$ at $x^\star = 2$, which is equivalent to mentally fitting a parabola to the four stimulus points and estimating the point of intersection with a vertical line at $x^\star$.

To train participants on the generative model and the prior, we showed participants the generative parabola after each trial. In the main experiment, we showed a set of 20 unique stimuli for each of the three noise levels $\sigma_g \in \{0.03, 0.1, 0.4\}$, and each unique stimulus was repeated 20 times. We denote the set of the 20 responses to the $j^{th}$ stimulus as $R_j = \{r_1^{(j)}, \dots r_{20}^{(j)}\}$. This amounts to a total of 400 trials per noise level. The order of the stimuli was randomized. For the additional experiment, we set $\sigma_g = 0.1$ and showed 10 unique stimuli 10 times. Fig 1 shows the experimental paradigm.

### Regression models

In each trial, we model the participant's computation by a consecutive inference and prediction step. During the inference step, the model assumes that the participant infers information about the quadratic parameter $w_j$ based on the presented data (i.e., stimulus) $D_j$. The inferred

information is then used for a subsequent prediction $y^\star$. We describe the participant's overall task as computing the *predictive distribution*: $p(y^\star|x^\star, D_j, \mathcal{M})$.

Prior regression (P-R) is our null model. P-R assumes that participants make predictions based on their prior belief but disregard information from the stimulus:

$$p(y^\star|x^\star, D_j, \mathcal{M}_{\mathrm{PR}}) = \int p(y^\star|x^\star, w)\pi(w)dw \qquad (6)$$

Maximum likelihood regression (ML-R) relies only on the likelihood maximizing parameter, $w_{ML}$:

$$
\begin{aligned}
p(y^\star|x^\star, D_j, \mathcal{M}_{\mathrm{ML}}) &= p(y^\star|x^\star, w_{\mathrm{ML}})\\
\text{with}\quad w_{\mathrm{ML}} &= \arg\max_w p(D_j|w)
\end{aligned}
\qquad (7)
$$

Maximum a posteriori regression (MAP-R) uses the parameter that maximizes the posterior $p(w|D_j) = p(D_j|w)\pi(w)/p(D_j)$:

$$
\begin{aligned}
p(y^\star|x^\star, D_j, \mathcal{M}_{\mathrm{MAP}}) &= p(y^\star|x^\star, w_{\mathrm{MAP}})\\
\text{with}\quad w_{\mathrm{MAP}} &= \arg\max_w p(w|D_j)
\end{aligned}
\qquad (8)
$$

Bayesian regression (B-R) uses the entire posterior for making predictions by marginalizing over it:

$$
\begin{aligned}
p(y^\star|x^\star, D_j, \mathcal{M}_{\mathrm{BR}}) &= \int p(y^\star|x^\star, w)p(w|D_j)dw\\
\text{with}\quad p(w|D_j) &= p(D_j|w)\pi(w)p^{-1}(D_j)
\end{aligned}
\qquad (9)
$$

Bayesian regression with noise estimate (B-R$_\sigma$) loosens the assumption that participants treat $\sigma_g$ as a hyperparameter and instead assumes they use an estimate $\hat{\sigma}_g$ on a trial-by-trial basis. Using the maximum likelihood estimator and the number of points $M = 4$:

$$\hat{\sigma}_g^2 = M^{-1}\sum_{i=1}^{M}(y_i - w^\star x_i^2)^2 \quad\text{with}\quad w^\star = \left(\sum_{i=1}^{M} x_i^4\right)^{-1}\sum_{i=1}^{M} y_i x_i^2. \qquad (10)$$

After substituting the estimate $\hat{\sigma}_g$ for the hyperparameter $\sigma_g$ in Eq (9), the posterior predictive distribution is computed analogously to B-R.

## Participants' internal noise

To predict the participants' responses $r$ from the regression models' output $y^\star$, we had to account for the internal noise of the participants. We did this by showing a noise-free stimulus 20 times and fitting a Gaussian with variance $\sigma_m^2$ to each participant's response distribution: $p(r|y^\star) = \mathcal{N}(r; y^\star, \sigma_m^2)$. To be robust against outliers, we took the average of the 16% and 84% percentiles of the response distribution as motor noise. The predicted response distribution is then

$$p(r|D_j, x^\star, \mathcal{M}) = \int p(r|y^\star)p(y^\star|x^\star, D_j, \mathcal{M})dy^\star \qquad (11)$$

## Model comparison

We used the log-likelihood and the variance to compare the predicted and empirical response distributions.

**Log likelihood.** To compute the log likelihood for a model $\mathcal{M}$ across all response at a given noise level $\sigma_g$, we summed the individual log likelihoods of each response $r$ (the log of Eq (11)) across all stimuli $D_j$:

$$\mathcal{L}_{\mathcal{M}} := \sum_{j=1}^{20} \sum_{r \in R_j} \log p(r|D_j, x^\star, \mathcal{M}) \tag{12}$$

**Bayesian random effects.** The winning model of the participant averaged log likelihood must not necessarily win the model comparison for each participant. The Bayesian random effects analysis quantifies what fraction of participants are described by a model [47]. Specifically, we report the expected likelihood of each model for a random participant (Eq. (15) in [47]), i.e., the normalised Dirichlet parameter: $\alpha_{\mathcal{M}}$.

**Variance prediction.** As a independent comparison of the data and the predicted response distribution, we used the variance of the responses. For each of the 20 stimuli $D_j$ we obtained a single empirical value from the 20 responses recorded:

$$\sigma^2_{r|D_j} = \frac{1}{19} \sum_{k=1}^{20} \left( \bar{r}^{(j)} - r_k^{(j)} \right)^2 \tag{13}$$

where high variance values reflect ambiguous and difficult stimuli while low values indicate easy stimuli, prompting participants to give very similar responses across repetitions. Hence, at each noise level $\sigma_g$, we have an empirical variance distribution that corresponds to the 20 stimuli $\{D_j\}_{j=1}^{2}0$.

For the predicted variance distribution, we use the variance predicted by a model $\mathcal{M}$ in response to a stimulus $D_j$:

$$\sigma^2_{r|D_j, \mathcal{M}} = \mathrm{Var}[r|D_j, x^\star, \mathcal{M}], \tag{14}$$

where we used Eq (3) for an analytical computation of the variance. To improve the resolution, we increased the number of stimulus samples $D_j$ to 5000 for the theoretical prediction. We use the resulting distribution over $\sigma^2_{r|D_j, \mathcal{M}}$ to compute the median in Fig 4 and the log density in the background.

## Determining the components of the variance in the response data

To compare the components of the predictive variance in Eq 3 to data, we make the following definitions for a set of response $\mathcal{R}_j$. The empirical mixing coefficient is the fraction of positive responses:

$$c = \frac{|\{r > 0|r \in \mathcal{R}_j\}|}{|\mathcal{R}_j|}$$

If only one of the modes is present in the data ($c \in \{0, 1\}$) the bimodal contribution vanishes and we do not require the means for the total variance. If both modes are present we compute their means:

$$\mu_1 = \mathbb{E}[r|r > 0] \qquad \mu_2 = \mathbb{E}[r|r < 0],$$

We define the unimodal variance contribution as the variance of the dominant mode:

$$\sigma_y^2 + \sigma_m^2 = \text{Var}[r|\tilde{\mathcal{R}}],$$

where $\tilde{\mathcal{R}}$ is the set of responses in the dominant mode, i.e., the mode that has the majority of responses. If no dominant mode exists we omit the stimulus. We did not use the inferior mode to have sufficient samples (at least 11) to estimate the variance.

### The unimodal dataset

To obtain a unimodal dataset from the full dataset, we consider only responses and model predictions if they have the same *dominant mode*, i.e., parabolas facing either upwards or downwards. We define the dominant response mode as the one containing more than half of the responses and the dominant mode of the model as the one carrying more than half of the probability mass. For example, if 11 responses fall into the upper mode but the models predict a downward parabola, all responses are disregarded. However, if the models predict an upward parabola the 11 responses enter the unimodal dataset. Note that the symmetric prior ensures that B-R, B-R$_\sigma$ and MAP-R share a dominant mode. Because the models use the same likelihood term, they process the stimulus as evidence for the same mode and break the symmetry in the same direction.

Averaged over participants, the fraction of trials per condition retained for the unimodal dataset is 0.994, 0.849, 0.596 and 0.703 for $\sigma_g \in \{0.03, 0.1, 0.4\}$ with $\sigma_\pi = 0.1$ and $\sigma_g = 0.1$, $\sigma_\pi = 0.5$, respectively.

### Participants

Seven naive participants (3 females, 4 males, ages 21-27) participated in the main experiment and four naive participants (all males, ages 21-30) took part in the second experiment. The experiments were programmed using custom software implemented in MATLAB. Stimuli were presented on a 1920x1080 (36 pixels/cm) monitor with a refresh rate of 120 Hz. Participants viewed the display binocularly. Each trial comprised a fixation dot presented for 1 s followed immediately by presentation of the stimulus (with 5 arcmin point diameter). Participants moved a red point up or down using the up and down arrow keys to indicate the vertical position of the parabola at the given horizontal location. See S1 Text for more details.

### Ethics statement

All participants gave informed consent in accordance with protocol 384/2011 "Commission cantonale d'éthique de la recherche sur l'être humain". Participants provided written consent prior to the experiment.

## Supporting information

**S1 Text. Derivations and additional details.**
(PDF)

**S1 Data. The file contains one folder for each participant $N \in \{1, \ldots 11\}$.** Within each folder, the name of the text file indicates the parameters. The participants 1...7 completed four conditions of generative noise $\sigma_g \in \{0, 0.03, 0.1, 0.4\}$ and the variance parameter was $\sigma_\pi = 0.1$. For example, the file subj1_sig_g = 0.1.txt contains all trials of the first participant with generative noise $\sigma_g = 0.1$. The participants 8...11 completed only one condition: $\sigma_g = 0.1$ and $\sigma_\pi = 0.5$. Since the variance parameter is different from its default value, we indicate it explicitely in the

file name, e.g. subj8_sig_pi = 0.5_sig_g = 0.1.txt. Each data file contains 11 columns. The first eight columns describe the $x$ and $y$ coordinates of the stimulus points. The last three columns contain (in that order) the stimulus index $j \in \{1, \ldots 20\}$, the generating quadratic parameter $w_j$ and the vertical location of the observed response.
(ZIP)

## Author Contributions

**Conceptualization:** Jannes Jegminat, Maya A. Jastrzębowska, Matthew V. Pachai, Michael H. Herzog, Jean-Pascal Pfister.

**Data curation:** Maya A. Jastrzębowska, Matthew V. Pachai.

**Formal analysis:** Jannes Jegminat.

**Investigation:** Jannes Jegminat, Maya A. Jastrzębowska, Matthew V. Pachai.

**Methodology:** Jannes Jegminat, Maya A. Jastrzębowska, Matthew V. Pachai, Michael H. Herzog, Jean-Pascal Pfister.

**Resources:** Michael H. Herzog, Jean-Pascal Pfister.

**Software:** Jannes Jegminat, Maya A. Jastrzębowska.

**Supervision:** Michael H. Herzog, Jean-Pascal Pfister.

**Visualization:** Jannes Jegminat.

**Writing – original draft:** Jannes Jegminat, Jean-Pascal Pfister.

**Writing – review & editing:** Jannes Jegminat, Maya A. Jastrzębowska, Michael H. Herzog, Jean-Pascal Pfister.

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
