## [Decision Letter · Decision Letter 0]

9 Sep 2019

Dear Dr Jegminat,

Thank you very much for submitting your manuscript 'Bayesian regression explains how human participants handle parameter uncertainty' for review by PLOS Computational Biology. Your manuscript has been evaluated by three independent peer reviewers and myself. As you will see in the comments below, the reviewers believed that the study addresses a relevant and interesting research question. However, they also raised substantial concerns about the manuscript as it currently stands. For example, Reviewer #1 is concerned about the framing of the paper and also wonders – just as Reviewer #2 –whether the work is as novel as presented. Also, both Reviewers #2 and #3 are concerned that the use of the bimodal distribution over w may have favored some models over others, which may have implications for the generalizability of the results and conclusions. Please see the reviews below for a more detailed comments.

While your manuscript cannot be accepted in its present form, we are willing to consider a revised version in which the issues raised by the reviewers have been adequately addressed. We cannot, of course, promise publication at that time.

Sincerely,

Ronald van den Berg

Associate Editor

PLOS Computational Biology

Samuel Gershman

Deputy Editor

PLOS Computational Biology

[LINK]

Reviewer's Responses to Questions

**Comments to the Authors:**

Editor: 

I have read the paper with great interest and have a few points in addition to the concerns raised by the reviewers. My main comment concerns that statement at the very end of Results that “The two additional variants of B-R perform slightly worse than the original B-R in terms of the log-likelihood”. Perhaps I misunderstood, but isn’t this in direct contradiction with the conclusion that the data support “..the idea that the generative noise is indeed jointly estimated with the quadratic parameters”? The BR_f model may provide a better fit to the variance curves (a summary statistic), but this apparently comes at the cost of not explaining other parts of the data as well as BR.

I agree with Reviewer #3 that the presentation of the model comparison results can be improved. I suggest to either go with the solution proposed by this reviewer (compute ratios w.r.t. baseline model) or go with relative values (subtract for each subject the log lh of the overall best model from the log lh off all models, such that the best model has by definition a \\delta log lh of 0).  Also, i would suggest the present these results using a bar graph rather than lines (Fig. 3) and add error bars that indicate variability across subjects. Since there are only 7 subjects, you may even want to consider reporting the individual model comparison results (possibly in Supplement if it would make the main figures to cluttered).

On p. 7 it is mentioned that “The data is pooled across participants”. I assume that this is only a presentation issue and that the models were still fitted to individual data sets? Please clarify. Also, *if* the models were fitted to pooled data, please justify this choice as that would be a bit strange.

Finally, in the introduction it is mentioned that “Most […] experimental studies can be cast into the problem of estimating a hidden quantity from sensory input”. It may be worth noting that we have found evidence for Bayesian inference in various perceptual tasks that go beyond simple estimation, including visual search (e.g. Ma et al. 2011, Nature Neuroscience; Stendgård & Van den Berg 2019), same/different discrimination (Van den Berg et al 2012, PNAS), and change detection (Keshvari et al 2012, PLoS One).

Minor:

I think that the abbreviation “DNNs” on p. 2 was never introduced

Sigma_m is in the supplement referred to as “motor noise”, but in the main text it is explained that it more broadly captures “internal sources of noise arising during neural processing, decision making and the execution of motor action”. I prefer the broader interpretation of sigma_m and believe it would be useful to add a reference to Drugowitsch 2016 (Neuron) and Beck, Ma, et al 2012 (Neuron) here.

Reviewer #1: The authors ask if human participants feature behavior that is better explained by Bayesian regression rather than simpler alternatives/heuristics. The specific feature of Bayesian regression is that predictions need to take into account the uncertainty in the inferred regression coefficients. The authors test this by providing subjects with noisy point-wise observations of a quadratic function, and ask for the participants' best guess when extrapolating the (unobserved) function to more extreme values. They demonstrate that the participants' extrapolations are best explained by a Bayesian regression model rather than alternative models, such as maximum likelihood or maximum a-posteriori. This is, to my knowledge, the first demonstration that humans are able to take account of their uncertainty in model parameters when making predictions. Thus, it is novel, original, and should be of interest to a wide audience.

Unfortunately, the current manuscript leaves multiple, potentially confounding questions unanswered:

First, the authors seem to implicitly assume that predictions are made by drawing samples from the posterior predictive density (after which some motor noise is added). This is just one possible way of turning posteriors into decisions. Bayesian decision theory posits that decisions ought to be made by picking an estimate that minimizes a particular expected loss under the posterior. The squared loss, for example, would result in deciding according to the posterior mean (see Bishop (2016) and Kording & Wolpert (2004)). There exists no loss function, however, that makes posterior sampling the optimal strategy. Therefore, Bayesian decision theory does not currently justify the performed analysis. One could suggest that the participants do not act according to Bayesian decision theory, and instead sample from this posterior. However, this does not conform to recent work (Drugowitsch et al. (2016)) showing that behavioral variability seems to arise from noisy inference rather than noisy decisions. Thus, I urge the authors to elaborate in more detail how they believe posteriors are turned into choices, and potentially investigate if noise in the inference itself could result in the behavior that they aim to describe.

Second, the model comparison results seem to be driven by the bimodality of participants' responses (i.e., convex vs. concave function), which appears only supported by Bayesian regression. That by itself would be an interesting result, but needs further investigation. If you would restrict responses to the single, dominant mode (e.g., ignoring all trials in which the other mode was chosen), would the Bayesian regression model still win the model comparison? Would a simpler model that only considers two possible functions (i.e., convex or concave?) + some motor noise perform worse than the Bayesian model?

Third, model comparison is performed on the mean log-likelihoods. Even though Fig. 3 might suggest otherwise, this doesn't per se exclude the possibility that the model comparison isn't driven by a few participants. I would suggest two modifications. First, the models should be compared in terms of log-likelihood ratios, usinng the preferred model as baseline. Then, all measures are relative to the model fit of this preferred model, and negative values indicate a worse model fit of alternative models. Averaging across those would correspond to a more informative within-subject comparison. Second, to exclude the possibility of few participants driving the model comparison I would suggest additionally performing a Bayesian random effects model comparison (see Stephan, Penny et al. (2009)).

More generally, the presentation of the results could be improved. Some statements are ambiguous or simply wrong (see below), and I would encourage the authors to carefully revise the manuscript to improve its clarity. The SI derivations are lengthy. They rely on partially known results and can be shortened (see below).

Detailed comments:

Abstract: "The quadratic parameter was drawn from a prior distribution, unknown to the observer" - what was unknown to the observer? The quadratic parameter? Or that this parameter was drawn from a prior distribution?

p3, A novel paradigm to test regression: I found this section generally confusing, and I encourage the authors to revise the presentation order to make stimuli and experiment clearer. I understand that you have 20 4-dot stimuli with the same underlying wj. For a fixed noise level, did they also have the exact same noise instantiation - i.e., was the stimulus the same on the screen? Were they the same across participants? Furthermore, you repeat each stimulus 20 times, which would imply 400 stimuli overall. How were these 400 stimuli distributed across the different blocks of different noise level? What about the 10 practice trials?

p4, Fig 2 caption: "Contours indicate the equiprobable responses predicted by ..." - not precise as responses only happen at x=2. Is this the predicted response distribution if subjects were asked to respond at different x's?

For (D), are those the response distributions corresponding to the dot pattern in (A)?

p5, "To complement the log-likelihood analysis [...], we used the variance" - which variance?

p6, Fig. 4 caption: "The errors of the other models are too small to show" - errors? Or variance? SEM? SD?

p9, "In each trial, the participants carried out [...]" - you only know that their behavior was compatible with the proposed computations, but not the exact computations participants performed to feature this behavior.

p9, "in Bayesian regression generalizes Eq (8" - ")" missing

p9, "as in ordinary B-R (Eq (8)" - ")" missing

p10, "(the log of Eq (10)" - ")" missing

Fig 4 B/C/D: please consider horizontal jitter of the grey dots instead of blurring them.

SI, general: many of the derivations are standard in the Bayesian literature, and their details can be skipped or they can be simplified (e.g., Bishop (2016), Appendix B, "Gaussian")

SI, p2: "see ??" - broken ref

SI, p3, "Fig 2(A) / Fig 2(B)" - should this refer to Fig. S2?

SI, p4, "[...] because our mixed Gaussian prior is the conjugate prior of the Gaussian likelihood" - "is A conjugate prior", as there are multiple.

SI, p6, "[...], we the posterior maintains" - the "we" should be removed

SI, p8, Derivation of full Bayesian regression: wouldn't a normal inverse-gamma mixture provide a conjugate prior for this case? This would lead to analytical posteriors.

Reviewer #2: The authors asked how humans handle parameter uncertainty in a regression task. Observers had to extrapolate from 4 points generated from a noisy quadratic function to predict the y-coordinate of a 5th point whose x-coordinate was given. They compared Bayesian regression to prior regression, ML regression, and MAP regression, and found the best performance for Bayesian regression models fit to the human data. The question is important, as function estimation is a basic and general task. The paper was very clearly written and technically proficient. The evaluation of the models by two separate methods (log likelihood and response variance) was a strength. All methods details were provided. However, I wondered whether the claim that humans use Bayesian regression depended critically on the specifics of the task and noise assumptions. I also wondered how this paper goes beyond previous arguments that humans use Bayesian regression.

Major comments:

1. Novelty: Lucas et al. 2015, Psychonomic Bulletin & Review, argue that previous models of how humans do function estimation can be recast as Bayesian regression. They also compare performance of various models and show that Gaussian process models, which are closely related to Bayesian regression models, fit human data well. How does the current work go beyond these previous arguments for Bayesian regression?

2. It seems that the MAP model failed primarily because it did not capture the bimodal distribution of observers’ responses. In this task, the parameter of interest, the coefficient on the quadratic term of the function, was drawn from a bimodal prior distribution, so that the parabola sometimes opened upward and sometimes opened downward. For the same 4-dot stimulus, human observers sometimes extrapolated for a downward parabola and sometimes for an upward parabola, whereas the MAP model always extrapolated in just one direction. But if I understood correctly, all models knew the generative noise, unlike the human observers. How much do the unimodal predictions of the MAP (and ML) models rely on this assumption? That is, if the noise were estimated trial by trial, would this stochasticity lead to bimodal predictions even for MAP and ML models? This concern seems especially relevant given that the authors show better performance for versions of the Bayesian model that estimate the generative noise trial by trial, and it seems that this possibility was not tested for the non-Bayesian models.

3. Relatedly, is there any aspect of the data that argues for Bayesian regression that is not dependent on the bimodal pattern of extrapolation reports? It seems that both the log likelihood results and the variance results were driven by the bimodal pattern, which raises the question of whether the result would generalize to a case where the prior is unimodal. This seems important, as unimodal priors on parameters are probably more typical in natural tasks.

4. The experiment consisted of 20 repetitions of each of 20 4-dot stimuli. Feedback was presented after every trial, which created the possibility that the observers could learn a direct mapping between a certain dot configuration and an extrapolated point over the course of the experiment. I wondered whether such learning could create the appearance of Bayesian performance, and more generally whether there was any evidence for learning over the course of the experiment.

5. The B-R3 model seemed a bit ad-hoc. Is there a theoretical reason for estimating the noise based on two sets of 3 points rather than all 4 points (the B-R4 model reported in the supplement, which did not perform as well)?

Minor comments:

1. The significance statement is quite vague. It seems like it could apply to almost any Bayesian modeling paper.

2. To motivate the study, it may be worth pointing out in the intro that while “many aspects of cognition are indeed well described by Bayesian statistics” (p. 2), many are not (Rahnev & Denison 2018).

3. “This feedback enabled the participants to learn both the prior and the generative model.” (p. 3) What is the evidence that these were learned successfully?

4. It may be helpful to explain why the MAP model does not predict a bimodal distribution even though it uses the prior.

5. For the variance analysis, why was the variance within a given stimulus (observers) compared to the variance generated across a large number of stimuli (models), rather than comparing variance from the same source?

6. For B-R3, how was the ML estimate of the generative noise obtained? (p. 9) Does the procedure assume that w is known?

7. Some figure references were missing or in a different format (e.g. “Bottom right” vs. “D”).

Reviewer #3: This is a very neat paper which made me wish I’d thought of doing that experiment. Essentially, the manuscript shows some reasonably convincing evidence that people can complete a simple regression problem in a Bayesian manner.

Major points

I felt that the manuscript treated the division between the inferential tasks of model comparison, regression, prediction, parameter estimation, posterior prediction, as too sharply divided. While these _concepts_ are distinct, once we start thinking about people completing real behavioural tasks, then I think it becomes a bit less clear to claim the completion of a task only involves one inferential concept and not others. I think the authors could make their claims more robust by adding more reasoning why this task should be considered as regression as opposed to prediction. Someone could make a reasonable case that the participants are doing posterior prediction here, so the case for regression could be firmed up somewhat. Someone may argue that regression is not a 'core' inferential task in that there is just model learning, parameter estimation, and posterior prediction. If so, then the paper demonstrates that people can learn a parabolic model and do parameter estimation and/or posterior prediction. I think the authors could perhaps update the manuscript to make their claims more robust to someone with these views.

Minor points

The model comparison is nice, but I think it would also be useful very early on to include a short intuitive statement about what kind of results would really support or disconfirm that people are doing this regression task in a Bayesian manner.

Clarify if participants were naive to the goals of the study.

This is touched upon in the manuscript, but it might be worth elaborating or speculating a little more about what might happen outside of a toy scenario where the participants know the data generating model. Presumably people would have to simultaneously conduct parameter estimation (of multiple models) and do inference over models.

Unless I've misunderstood, a core part of this manuscript is the bimodality of the parameter distribution. I felt that this was not given much attention in the manuscript so perhaps there is some scope for clarification.

Along the lines of the previous point, I was a bit unsure how to interpret the density plot for B-R in figure 2-A. The methods spell out that the mixture of up and downwards parabolas was 50%. This is not the case in this density plot, so presumably this plot is the best fit to a given participant? Perhaps this point of confusion on my part can help the authors with clarifications in the main text and/or figure legend.

Possibly a pedantic point, but Panels B-D in Figure D could feasibly be rotated so the y coordinate is on the y axis, to make it a little more intuitive. But this is just a suggestion for consideration.

**Have all data underlying the figures and results presented in the manuscript been provided?**

Reviewer #1: Yes

Reviewer #2: Yes

Reviewer #3: Yes

PLOS authors have the option to publish the peer review history of their article (what does this mean?). If published, this will include your full peer review and any attached files.

Reviewer #1: No

Reviewer #2: No

Reviewer #3: No

---

## [Decision Letter · Decision Letter 1]

17 Jan 2020

Dear Mr. Jegminat,

Thank you very much for submitting your manuscript "Bayesian regression explains how human participants handle parameter uncertainty" for consideration at PLOS Computational Biology. Your manuscript was reviewed myself and the three original independent reviewers. As you will see in the reviews attached below, no major concerns were raised, but there were still a few minor concerns.

Reviewer #1 believes that the writing is too strong at several places and somewhat imprecise at other places. For example, in the abstract your write "Our results show that humans use Bayesian regression.". I agree with the reviewer that it would be good to soften statements like this one, because this is much stronger than what is justified by the results. Moreover, Reviewer #2 has a suggestion for a minor addition or change in the analyses.

Since the remaining issues are minor, I will probably move to a decision without sending the manuscript back to external reviewers.

Sincerely,

Ronald van den Berg

Associate Editor

PLOS Computational Biology

Samuel Gershman

Deputy Editor

PLOS Computational Biology

[LINK]

Reviewer's Responses to Questions

**Comments to the Authors:**

Reviewer #1: I appreciate the authors' changes to the manuscript. I think that the model comparison, as well as the analysis of the response variance is much improved. I don't have any major comments on the details of the analysis and its description. Two issues nonetheless remain, as I will describe in turn.

The first is that not all statements in the manuscript are immediately supported by the performed analyses. This is a thread that goes through the whole manuscript, such that pointing out every single instance would be too tedious. Some of the most glaring instances are:

* Probability matching/sampling: the manuscript states that human participants perform sampling/probability matching, but the authors haven't tested alternative models of noisy inference. The latter requiring to fit parameters might be a reason for the authors to not want to test them, but it is not a reason for ruling out the possibility that the observed variability is due to noisy inference. Thus, all statements that state that humans choose by probability matching/sampling needs to be toned down/relativized. This starts with the abstract ("We further add evidence that humans use probability matching rather than Bayesian decision theory..." - a noisy inference model might explain the data equally well), over the Results (pages 6/7), to the Discussion. The discussion at least mentions the possibility of noisy inference as an open question, but the rest of the manuscript should word its conclusions more carefully - in particular in the light of increasing evidence (e.g., from Wyart lab) that behavioral variability that has been previously ascribed to posterior sampling can be equally well (or better) explained by noisy inference. Thus, strongly stating that participants sampled might, in fact, be wrong.

* Abstract: "Our results show that humans use Bayesian regression" - they suggest, but don't show

* p8: "An explanation for this discrepancy is that participants estimated the noise on a trial-by-trial basis" - this sounds as if you _knew_ that participants performed this estimation. Wouldn't it be better to call this "A potential explanation"?

The second is an overall lack of precision in parts of the manuscript. There are multiple ambiguous/imprecise statements, and leaving it up to the reader to deal with them might lead to misunderstandings. Thus, I urge the authors to resolve any potential ambiguities in their writing. Examples include:

* p3: "The proposal is to apply Bayesian regression to DDNs." - who proposes this? Are you? Do others?

* p3: "The presentation order was randomized" - this is imprecise. It was randomized within each noise level, but blocked across noise levels.

* p3: "The advantage of a response distribution over single responses" - this is contradictory, as a single response doesn't give a distribution. Should this be "The advantage of observing several responses for the same exact stimulus is that we can compare the set of responses to the response distribution predicted by the different models"?

* p5: "[...] motor noise is the only source of uncertainty" - do you mean that it is the only source of response variability? Uncertainty is never measured.

* p7: "we use the model likelihood p(M|Rk)" - p(M|Rk) is the model posterior, not the model likelihood, which would be p(Rk|M).

* p7: "shows that the majority of the population" - do you mean that the majority of the participants' responses?

* p7: "the transition from unimodality to bimodality" - of what? The response distribution? The model posterior?

* p7: "we use the response variance" - which variance? The variance of responses in trials in which the same stimulus was shown? Please be precise.

* p12, "regression and inference differ" - too general use of "regression" and "inference" (in particular, as regression is a form of inference), please relativize, e.g. "Algorithmically, however, linear regression and Gaussian Process inference differ" - also for later instances of "regression" and "inference" in this paragraph.

* p15, "if they have the same dominant mode, i.e. either up or down" - too colloquial and imprecise.

Additional details:

p3: "In our main experiment, we set the standard deviation of each prior model to sigma_pi = 0.1" - this is unclear, as sigma_pi hasn't been defined. Overall, this section would benefit from adding the stimulus equations (4) and (5). This would make it significantly easier to understand the generation of the stimulus. "We fixed the values of x positions ..." - do you want to say that those are the locations where the dots are shown?

p4, footnote 1 - shouldn't that footnote appear earlier, when you introduce the MAP-R model?

p5, "Fig 2(E) illustrates that participant's responses cover a wide range of values" - would be useful to state upfront that Fig 2(E) relates to increased sig_pi.

p6, Fig. 3 "Negative value favour sampling" - should be "valueS"

p8, "because because" - repetition

p11, "It is an interesting question whether subjects solve regression tasks with model uncertainty by taking advantage ..." - resolving model uncertainty has previously been studied under the name of "causal inference". See, for example, Koerding et al. (2007), PLoS ONE.

p12, "through constant feedback" - "constant feedback" makes it sound as if the feedback was the same across trials. "continuous feedback", or "feedback in every trial" might be better.

Supplement page 8: "... we disregarded that model" - "disregard" sounds as if you don't consider this model to be valid. Please rephrase to state that you don't consider it in the model comparison, but that it is nonetheless a valid model.

Fig. 2A: please change colors to make sure that the 4 dots are visible.

Fig. 4: green line is not visible

Reviewer #2: The authors have revised the manuscript thoroughly and addressed all of my comments. The addition of the unimodal analysis has strengthened the paper. Overall, this is a clear and thoughtful manuscript addressing an important question.

I have one concern about the new learning analyses: I am not sure it is appropriate to include all noise conditions in a single regression (Figure S3). Different amounts of learning could be expected for the different noise conditions, because there is more ambiguity for high noise, and thus observers could benefit more from learning those specific stimuli across trials. This does seem to be the case in the data – sigma_g = 0.1 and 0.4 data points fall slightly below the unity line. I agree with the authors that the magnitude of learning is small and is unlikely to be a serious methodological concern (also considering Figure S4). Still, I would recommend performing separate statistical tests for the different noise conditions and, depending on the outcome, perhaps modify the claim in the manuscript of no learning.

Reviewer #3: Major point: The authors adequately respond to this point, with updated text on page 12.

Minor points: The authors have given attentive responses to all my minor points.

I have read the revised manuscript - I think that this version is significantly improved compared to the original manuscript. Overall, I think this is a rigorous study which the readers of PLoS Computational Biology will be interested in reading.

**Have all data underlying the figures and results presented in the manuscript been provided?**

Reviewer #1: Yes

Reviewer #2: None

Reviewer #3: None

PLOS authors have the option to publish the peer review history of their article (what does this mean?). If published, this will include your full peer review and any attached files.

Reviewer #1: No

Reviewer #2: No

Reviewer #3: Yes: Dr Benjamin T. Vincent
---

## [Editor Report · Decision Letter 2]

19 Apr 2020

Dear Mr. Jegminat,

My apologies for the slight delay in handling the revision of your manuscript 'Bayesian regression explains how human participants handle parameter uncertainty'. Since the revision was rather minor, I decided to not send it back to the reviewers, as I indicated in my previous decision letter. I believe that you adequately addressed all remaining issues that were raised by the reviewers and am pleased to inform you that your manuscript has been provisionally accepted for publication in PLOS Computational Biology. Please note that there is a small typo in one of the changes you made ("to what extend" should be "to what extent").

Best regards,

Ronald van den Berg

Associate Editor

PLOS Computational Biology

Samuel Gershman

Deputy Editor

PLOS Computational Biology

---

## [Editor Report · Acceptance letter]

11 May 2020

PCOMPBIOL-D-19-01115R2 

Bayesian regression explains how human participants handle parameter uncertainty

Dear Dr Jegminat,

I am pleased to inform you that your manuscript has been formally accepted for publication in PLOS Computational Biology. Your manuscript is now with our production department and you will be notified of the publication date in due course.

With kind regards,

Matt Lyles
